# Navigating Multidimensional Ideologies with Reddit's Political Compass: Economic Conflict and Social Affinity

## ABSTRACT

The prevalent perspective in quantitative research on opinion dynamics flattens the landscape of the online political discourse into a traditional left–right dichotomy. While this approach helps simplify the analysis and modeling effort, it also neglects the intrinsic multidimensional richness of ideologies. In this study, we analyze social interactions on Reddit, under the lens of a multi-dimensional ideological framework: the political compass. We examine over 8 million comments posted on the subreddits /r/PoliticalCompass and /r/PoliticalCompassMemes during 2020–2022. By leveraging their self-declarations, we disentangle the ideological dimensions of users into economic (left–right) and social (libertarian–authoritarian) axes. In addition, we characterize users by their demographic attributes (age, gender, and affluence).

We find significant homophily for interactions along the social axis of the political compass and demographic attributes. Compared to a null model, interactions among individuals of similar ideology surpass expectations by 6%. In contrast, we uncover a significant heterophily along the economic axis: left/right interactions exceed expectations by 10%. Furthermore, heterophilic interactions are characterized by a higher language toxicity than homophilic interactions, which hints at a conflictual discourse between every opposite ideology. Our results help reconcile apparent contradictions in recent literature, which found a superposition of homophilic and heterophilic interactions in online political discussions. By disentangling such interactions into the economic and social axes we pave the way for a deeper understanding of opinion dynamics on social media.

## CCS CONCEPTS

• **Applied computing** → **Sociology**; • **Human-centered computing** → **Social networks**; • **Information systems** → **Social networks**.

## KEYWORDS

Homophily, Polarization, Socio-Demographic, Reddit

**ACM Reference Format:**
Anonymous Author(s). 2018. Navigating Multidimensional Ideologies with Reddit's Political Compass: Economic Conflict and Social Affinity. In *Proceedings of The Web Conference (WWW '24)*. ACM, New York, NY, USA, 13 pages. https://doi.org/XXXXXXX.XXXXXXX

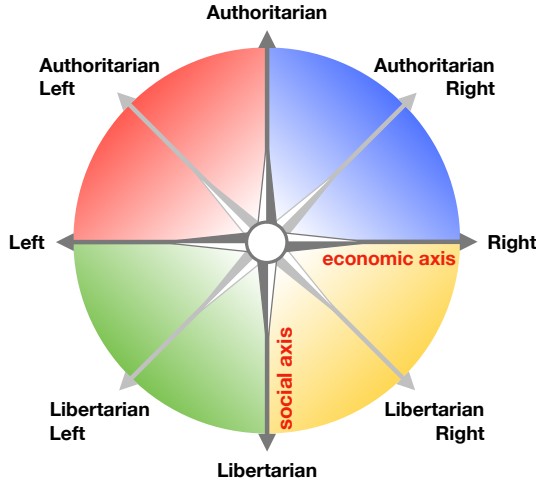

**Figure 1: The political compass [41]: the horizontal axis delineates the economic ideologies, moving from left (equality-focused) to right (liberty-focused), and represents views on resource allocation. The vertical axis delineates the social ideologies, from libertarian at the bottom to authoritarian at the top, and represents views on personal freedom.**

## 1 INTRODUCTION AND BACKGROUND

Understanding the dynamics of opinion formation in a population is a goal shared by researchers from different disciplines, from social and computer science to physics. Numerous studies have revealed the presence of opinion polarization in political discussions [43]: The phenomenon whereby two distinct groups tend to have opposite and potentially extreme views on a specific controversial topic, spanning religion, race, climate change, political ideology, and more [32, 38, 44, 55, 64].

When social interactions among individuals are taken into account, we often observe value homophily: individuals prefer to interact with peers that hold similar opinions [14, 15, 32]. The combination of opinion polarization and homophilic interactions leads to *echo chambers* [13, 33], a situation where existing beliefs can be reinforced by exposure to similar opinions. Echo chambers, in turn, contribute to opinion polarization by reinforcing ideological separation and strengthening the social identity of opposing groups [21, 34, 45]. These phenomena are easy to observe on social media such as Facebook or Twitter, where people share their opinions in more informal settings [2, 10, 14, 29, 32].

Other studies, however, find contrasting results [3, 23, 35, 60]. For instance, researchers have observed heterophilic interactions between Clinton and Trump supporters on Reddit, a preference for cross-cutting political interactions that contradicts the echo chamber narrative [18]. Likewise, some scholars attribute opinion polarization to demographic and socioeconomic factors rather than to social media [24, 46, 58]. Indeed, the profound disparities

and imbalances found in our society, such as by gender, race, age, and affluence, have become more closely tied to party affiliation and ideological stances [11, 51, 57]. This phenomenon, known as partisan sorting [42], may also be one of the causes of political polarization [60].

The answer to this apparent contradiction may lie in recognizing the intrinsic multidimensional nature of opinion dynamics. Indeed, the process of opinion formation builds upon views on multiple topics, as they might be discussed at the same time. When considering multiple topics, an interesting phenomenon frequently observed is issue alignment, i.e., the presence of correlations between opinions on different topics, particularly along the left–right dimension [26, 28]. For instance, individuals with strong religious beliefs tend to oppose abortion legalization [1], and various other non-trivial correlations manifest [4, 9, 22]. However, the prevalent view in quantitative research on opinion dynamics has focused on the simplest case of one-dimensional opinions concerning a single topic, both at the level of the analysis [5, 14] and modeling efforts [7, 16, 19, 36]. Only recently researchers have started taking into account a comprehensive multidimensional modeling framework for opinion dynamics [8, 12, 56].

Given these considerations, the main research question of this work is: *"How do interactions on social media align with respect to a fine-grained characterization of users in terms of their ideological coordinates and demographic features?"* In addition, we explore how conflictual interactions across and within ideological groups are.

To fulfill this research goal, we disentangle the social interactions of Reddit users along two ideological axes of the political compass [39] as depicted in Figure 1: one for economic values and resource allocation (left–right), and one for social values and personal freedom (libertarian–authoritarian). We take the multidimensional political ideology of each user from self-declared tags associated with active accounts on the `/r/PoliticalCompass` and `/r/PoliticalCompassMemes` subreddits during the period 2020–2022. This approach allows us to avoid inference techniques and achieve greater accuracy for the identification of the ideology of users. We infer socio-demographic features (age, gender, affluence, and partisanship) and relate them to the ideological orientation of users. Our results show that libertarians are older, richer, and more left-wing than authoritarians, while those on the left are younger, have a higher female representation, and are generally less wealthy than right-wing users.

Then, we examine the interactions among users along the two axes and according to their demographics. Comparing them to those obtained from a null model of interactions, our empirical observations show marked homophily on the social axis and for demographic characteristics. Conversely, the interactions are more heterophilic than expected on the economic axis. Finally, by analyzing the toxicity of the language associated with the interactions, we find that ideological cross-group interactions present higher-than-expected toxicity. Within-group interactions, on the contrary, show toxicity levels lower than the one expected from the null model, hinting at a certain degree of social affinity [62]. Overall, this multidimensional approach allows us to reconcile the apparent contradictions observed in the literature, particularly on the superposition of heterophily and homophily in online political discussions.

The main contributions of this paper are as follows:

- We examine a large dataset of political interactions on social media where users provide self-declared ideological positions on the political compass, and characterize it both in terms of ideological and demographic composition (Section 2);
- We study the political discussions occurring in this dataset by reconstructing the social interaction network among users, and discover that interactions along the social axis (libertarian–authoritarian) are more homophilic than expected (and similarly for demographic features), while interactions along the economic axis (left–right) are more heterophilic than expected (Section 3);
- We analyze the textual content of these interactions, and find that cross-group interactions are characterized by a higher level of toxicity, which hints at a conflictual discourse between axes' poles (Section 4).

The main body of the paper describes the results of our analysis for `/r/PoliticalCompass`. The ones for `/r/PoliticalCompassMemes` are qualitetively and quantitatively similar and are shown in the Appendix. To allow for reproducibility our code is available online.[1]

## 2 IDEOLOGY AND DEMOGRAPHICS OF USERS

### 2.1 Ideological Features

**Reddit's Political Compass** is a community that self-describes as "A subreddit for posting and discussing test results as well as political self-tests and political theory". On Reddit, *flairs* are tags that users add to their usernames or posts for additional context or to signal their identity. In the `/r/PoliticalCompass` (`/r/PC`) and `/r/PoliticalCompassMemes` (`/r/PCM`) subreddits, these flairs act as self-declarations and specify the political ideologies of users. Flairs are organized along the two axes shown in Figure 1: economical (left, center, right) and social (libertarian, center, authoritarian).

Using the PushShift API,[2] we collect comments and posts from `/r/PC` and `/r/PCM`. Henceforth, we analyze a dataset covering the period from 2020 to 2022, which is the most active one, see Figure A.1 in the Appendix. The dataset comprises 79 368 posts and 952 550 comments for `/r/PC` and 383 169 posts and 22 653 346 comments for `/r/PCM`, respectively. These represent 95% and 96% of the entire content in `/r/PC`, and 96% and 98% in `/r/PCM`.[3]

For our analysis, we select real active users identified with a unique political ideology, thereby filtering out potential bots [54], deleted accounts, those not affiliated with any political ideology or affiliated to multiple ideologies.[4] The breakdown of posts and comments by these user types is detailed in Table A.1.

**Economic Axis.** The economic or distributive axis measures possible opinions of how people should be endowed with resources. The *left* (equality) pole is defined as the view that assets should be

---

[1]https://anonymous.4open.science/r/political-compass-D741

[2]https://github.com/pushshift/api

[3]Our decision to focus on the 2020-2022 time frame is justified by this high coverage. Furthermore, while the `/r/PC` subreddit has been active since 2012, political ideologies have become a prominent feature only since November 2019.

[4]Within the `/r/PC` dataset for 2020-2022, 22 503 users are aligned with a single ideology, whereas 2544 have multiple ideological affiliations. In `/r/PCM`, these numbers are 258 428 and 30 658 respectively. We defer the analysis of users with multiple ideological declarations to a further study.

redistributed by a cooperative collective agency: the state in the socialist tradition, or a network of communes in the libertarian or anarchist tradition. The *right* (liberty) pole is defined as the view that the economy should be left to the market system, to voluntarily competing individuals and organizations. This is the classical left–right conflict that dominated the Cold War [39–41, 50].

**Social Axis.** The other axis—cross-cutting the first one—is concerned with values of fraternity, understood as axiological principles driving institutionalization, community, forms and actors of democracy, and the quality of the process of collective outcomes. This dimension measures possible political opinions either in a communitarian or procedural sense, considering the appropriate amount of personal freedom and participation: *libertarianism* is defined as the idea that personal freedom as well as voluntary and equal participation should be maximized. This would entail the full realization of liberty in a democratic sense. Parts of that view are ideas like autonomous, direct democratic institutions beyond the state and market, the transformation of gender roles, and self-determination over traditional and religious orders. On the opposite end of the axis, *authoritarianism* is defined as the belief that authority and religious or secular traditions should be complied with. Equal participation and a free choice of personal behavior are rejected as being against human nature, or against necessary hierarchies for a stable society [39–41, 50].

**Collected data is representative of all classes.** Figure 2 shows a comprehensive overview of user engagement and content distribution across various political ideologies for /r/PC (Figure B.3 for /r/PCM). This includes a breakdown of the number of users and comments classified by political ideology. The goal is to provide a perspective on the ideological composition and activity within the community. The plot shows that the collected data is representative of all classes, in terms both of users and activity. Additionally, Figure A.1 in the Appendix displays the temporal distribution of posts and comments. It shows a marked increase starting from 2020, especially considering the ones authored by users with an ideological flair (self-declaration).

**Ethics statement.** This work follows the guidelines and the ethical considerations by Eysenbach and Till [25], Moreno et al. [47], Ramírez-Cifuentes et al. [52]. All the results provide aggregated estimates and do not include any information that focuses on single authors. The users in our study were fully aware of the public nature and free accessibility of the content they posted since the subreddits are of public domain, are not password-protected, and have thousands of active subscribers. Reddit's pseudonymous accounts make the retrieval of the true identity of users unlikely, therefore our research did not require informed consent. Since this research does not involve interventions, no approval was required by our Institutional Review Board.

## 2.2 Demographic Features

**Inferring socio-demographics.** To infer the socio-demographic characteristics of Reddit users—including age (old/young), gender (male/female), affluence (poor/rich), and partisanship (left/right)—we employ the unsupervised representation learning approach by Waller and Anderson [63]. This method assigns to each subreddit

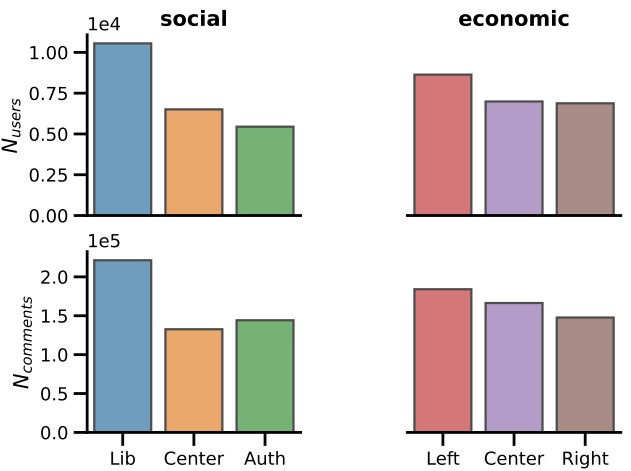

**Figure 2: Composition of `/r/PC` in terms of the number of users and comments classified by political ideology.**

a z-score for each of these four characteristics.[5] To determine the z-score for each user $u$ based on a characteristic $c$ from the set {age, gender, affluence, partisanship}, $z_u^{(c)}$, we compute a weighted mean of the z-scores $z_s^{(c)}$ of all subreddits $s \in S$. The weight is determined by the number of comments $N_{u,s}$ that user $u$ posted in subreddit $s$. The weighted average is thus given by:

$$z_u^{(c)} = \frac{\sum_{s \in S} N_{u,s} z_s^{(c)}}{\sum_{s \in S} N_{u,s}}. \tag{1}$$

Subsequently, we normalize the z-scores of users by using quantiles for each characteristic. These normalized scores will henceforth be referred to as *quantile scores* ($Q$). Quantile scores in the top 25% are classified as "high", those in the bottom 25% as "low", and quantile scores in between belong to the reference class. For instance, within the age characteristic, a high score would correspond to an "old" user, while a low score represents a "young" one. We stress that a score indicating an old-leaning user does not imply that such a user is necessarily old, as Reddit is known to be participated by more young than old users [6]. Rather, it indicates a user who frequents subreddits more likely to be participated by older users. A similar argument holds for the other categories: demographic attributes are always to be considered relative to the Reddit user base, and never on an absolute basis.

**Political declarations align with inferred ideologies.** Figure 3 illustrates the distribution of demographic characteristics among different political ideologies for /r/PC. In each plot, the y-axis represents the average quantile score of a demographic characteristic for users with the corresponding ideology on the x-axis. Notably, the distribution of inferred ideologies (left/right) closely mirrors the declared ideologies from the Political Compass along the economic axis, lending validation to our analytical approach. Exactly the same holds for /r/PCM, as depicted in Figure B.4.

---

[5]The study by Waller and Anderson [63] uses the 10k most popular subreddits from 2004-2018. Since our dataset begins in 2020, we focus on the overlap between our dataset and theirs, with a coverage of 57% in /r/PC and 68% in /r/PCM. As a result, socio-demographic characteristics could not be determined for 4% of users in /r/PC and 5% in /r/PCM.

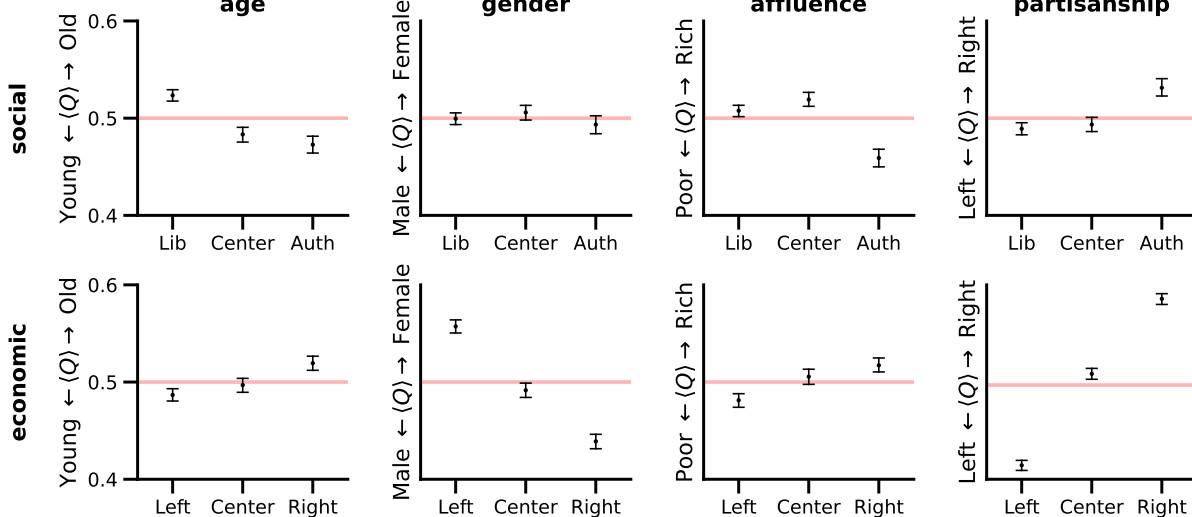

**Figure 3: For each axis (social and economic) displayed in the rows, and for each user characteristic (age, gender, affluence, partisanship) displayed in the columns, the plot shows the average quantile score and its 95% confidence intervals for /r/PC. Libertarians tend to be older, while right-leaning users are predominantly male. Left-leaning users are more likely to be female. Additionally, a correlation between partisanship and the left–right economic axis emerges, which further validates our data collection methodology.**

**Authoritarian and left-wing users are younger.** Figure 3 reveals noticeable variations in the characteristics of Reddit users based on their diverse socio-demographic attributes and political ideologies on /r/PC. Libertarians are typically older, wealthier, and more left-leaning compared to authoritarians. On the contrary, left-wing users tend to be younger and have a higher female representation but are generally less rich compared to right-wing users. Such results hold also for /r/PCM, as shown in Figure B.4.

## 3 HOMOPHILY AND HETEROPHILY IN SOCIAL INTERACTIONS

This section showcases the results regarding interactions among users grouped by political ideology and demographics. We start by describing the reconstruction of the interaction network, and the levels of homophily and heterophily among the distinct groups. Table 1 provides summary statistics for these networks.

### 3.1 Reconstructing the Interaction Network

The interaction network is represented as a directed weighted graph $G = (V, E, w)$, where users are nodes $V$ and the edges $E$ correspond to interactions between them. An edge $(u, v) \in E$ indicates that user $u$ (source) replied to user $v$ (target) in a thread on Reddit. Each edge carries a weight $w_{uv}$ which reflects the number of interactions between $u$ and $v$.

**Table 1: Network statistics: number of users $|V|$, edges $|E|$, average degree $\langle d \rangle$, and total number of interactions $W$.**

| Subreddit | $|V|$ | $|E|$ | $\langle d \rangle$ | $W$ |
|---|---|---|---|---|
| /r/PC | 18 135 | 173 672 | 9.58 | 261 078 |
| /r/PCM | 215 111 | 6 197 901 | 28.81 | 8 065 395 |

**Interaction Probabilities**. For a user with ideology $X_s$ from {libertarian, center, authoritarian} ($\{B, C, A\}$ for simplicity) on the social axis ($s$) and $X_e$ from {left, center, right} ($\{L, C, R\}$) on the economic axis ($e$), the probability of observing them is given by $P(u = X) = N_X/|V|$, where $|V|$ is the total number of users, and $N_X$ stands for those users identified with the ideology $X$. In particular, for /r/PC, the probability to find a node labelled as $X_s \in \{B, C, A\}$ (or $X_e \in \{L, C, R\}$), is respectively $P(B) \simeq 0.48$, $P(C) \simeq 0.30$, $P(A) \simeq 0.23$ (or $P(L) \simeq 0.31$, $P(C) \simeq 0.38$, $P(R) \simeq 0.31$).

**Joint probabilities.** When considering a specific ideological axis, either social or economic, the joint interaction probability between a user of ideology $X$ and another of ideology $Y$ is $P(X \to Y) = \frac{W_{X \to Y}}{W}$, where $W$ is the total number of interactions in the network, and

$$W_{X \to Y} = \sum_{u,v \in V : u=X \wedge v=Y} w_{u,v}$$

is the total weight of directed edges from $X$ to $Y$. Empirically, in /r/PC we find:

$$P(X_s \to Y_s) \simeq \begin{array}{c} B \\ C \\ A \end{array} \begin{pmatrix} \begin{array}{ccc} B & C & A \end{array} \\ 0.21 & 0.11 & 0.13 \\ 0.11 & 0.08 & 0.08 \\ 0.12 & 0.07 & 0.10 \end{pmatrix}, \quad (2)$$

$$P(X_e \to Y_e) \simeq \begin{array}{c} L \\ C \\ R \end{array} \begin{pmatrix} \begin{array}{ccc} L & C & R \end{array} \\ 0.13 & 0.12 & 0.12 \\ 0.12 & 0.12 & 0.09 \\ 0.12 & 0.09 & 0.08 \end{pmatrix}. \quad (3)$$

**Conditional probabilities.** Let $W_{X \to} = \sum_Y W_{X \to Y}$ be the number of interactions generated from $X$, and $W_{\to Y} = \sum_X W_{X \to Y}$ those received by $Y$. To take into account the volume of interactions

from the different groups (e.g., $P(A \rightarrow) \simeq 0.29 < P(A) \simeq 0.23$, $P(B \rightarrow) \simeq 0.45 < P(B) \simeq 0.48$), we consider the conditional probability of getting an interaction $X \rightarrow Y$ given a source with ideology $X$

$$P(X \rightarrow Y \mid X) = \frac{P(X \rightarrow Y)}{P(X \rightarrow)} = \frac{W_{X \rightarrow Y}}{W_{X \rightarrow}}. \quad (4)$$

Empirically, for /r/PC we have

$$P(X_s \rightarrow Y_s \mid X_s) \simeq \begin{matrix} & B & C & A \\ B \\ C \\ A \end{matrix} \begin{pmatrix} 0.47 & 0.25 & 0.28 \\ 0.42 & 0.28 & 0.29 \\ 0.41 & 0.25 & 0.34 \end{pmatrix}, \quad (5)$$

$$P(X_e \rightarrow Y_e \mid X_e) \simeq \begin{matrix} & L & C & R \\ L \\ C \\ R \end{matrix} \begin{pmatrix} 0.36 & 0.32 & 0.32 \\ 0.37 & 0.35 & 0.27 \\ 0.42 & 0.29 & 0.29 \end{pmatrix}. \quad (6)$$

If people interact with each other irrespective of their ideology, we would expect everyone to have an equal chance of interacting with each group, depending only on the group size. However, we observe that this is not the case. Instead, Equations (5) and (6) show important deviations from this expectation, in addition to differences between the two ideological axes.

On the social axis, the interactions tend to be more homophilic than heterophilic (the on-diagonal entry has a larger weight in each column). This fact is especially evident between libertarians and authoritarians: authoritarians engage more with authoritarians (34%) compared to how much libertarians do (28%), and a similar pattern is observed in the opposite direction for libertarians (47% vs. 41%). Centrists also show reduced interactions with both groups and favor instead their own group.

Surprisingly, on the economic axis the trend inverts. There is a noticeable heterophily between left and right users (the off-diagonal entries have a larger weight in the respective columns). Left users receive more interaction from right ones (42%) than from within their group (36%), and the opposite holds true for right users (29% from the right vs. 32% from the left).

## 3.2 Null model of Social Network

To determine whether the patterns observed in the previous section are statistically sound, we need to compare the empirical interaction network with a null model that ignores political ideologies. We use a configuration model, which is a directed, weighted random network (RN) that preserves the in and out-degree sequences but rewires the connections among them. This way, we ensure that the political ideology of users does not affect their interactions.

**Conditional probabilities in the null model.** The conditional probability of getting an interaction $X \rightarrow Y$, given the source's ideology $X$ in the null model is independent of the class $X$ of the source. Instead, it depends solely on the in-degree of the target's class $Y$

$$P_{RN}(X \rightarrow Y \mid X) = \frac{P_{RN}(X \rightarrow Y)}{P(X \rightarrow)} = \frac{W_{\rightarrow Y}}{W}. \quad (7)$$

For the /r/PC dataset, the conditional probabilities for the social and economic axes are

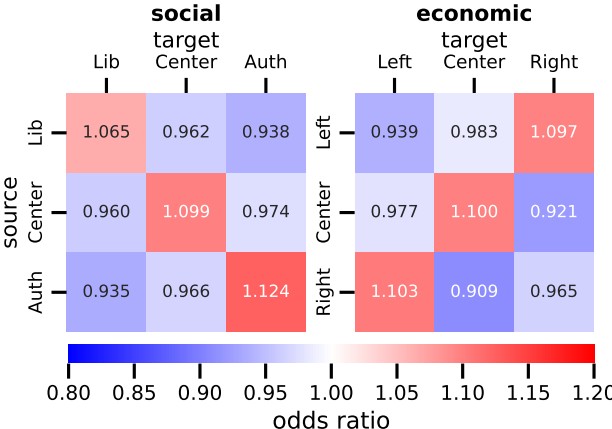

**Figure 4: Odds ratios between empirical and random conditional probabilities of interaction for /r/PC, with respect to social (left) and economic (right) axes. The interactions show a homophilic pattern on the social axis (higher values in the main diagonal) and a heterophilic one on the economic axis (higher values in the anti-diagonal).**

$$P_{RN}(X_s \rightarrow Y_s \mid X_s) \simeq \begin{matrix} & B & C & A \\ B \\ C \\ A \end{matrix} \begin{pmatrix} 0.44 & 0.26 & 0.30 \\ 0.44 & 0.26 & 0.30 \\ 0.44 & 0.26 & 0.30 \end{pmatrix}, \quad (8)$$

$$P_{RN}(X_e \rightarrow Y_e \mid X_e) \simeq \begin{matrix} & L & C & R \\ L \\ C \\ R \end{matrix} \begin{pmatrix} 0.38 & 0.32 & 0.30 \\ 0.38 & 0.32 & 0.30 \\ 0.38 & 0.32 & 0.30 \end{pmatrix}. \quad (9)$$

## 3.3 Effect of Ideologies

**Homophily within Social Ideologies.** Figure 4 illustrates that political ideologies on the social axis (libertarian–authoritarian) for /r/PC exhibit significant homophily. Interactions within the same ideology (on-diagonal) are up to 12% more likely than those predicted by the null model. The interaction probabilities both in receiving and sending a comment from/to a differing ideology are approximately symmetric. This pattern holds for /r/PCM too, as shown in Figure B.5, where interactions between authoritarian users are up to 18% more likely than expected.

**Heterophily across Economic Ideologies.** Conversely, the right side of Figure 4 shows a pronounced heterophily across ideologies on the economic axes. Left- and right-leaning users are approximately 10% more likely to interact with each other than what the null model predicts. This phenomenon of increased interaction between left and right is even more prominent in the /r/PCM subreddit, where is observed 15% more likely than predicted, see Figure B.5. Furthermore, within-group interactions for left and right are considerably lower than expected, while the economic center group shows more within-group interactions.

## 3.4 Effect of Demographics

Next, we test how social interactions are associated with demographic factors. To this aim, we employ the logistic regression model by Monti et al. [46] which outputs the probability of interaction between demographic groups and validates the statistical significance of the results.

**Logistic regression model.** We train the model for a link prediction task over the directed interaction graph $G = (V, E)$ discussed in Section 3.1. For a given node pair $u, v \in V$ the *target variable* is:

$$y_{u,v} = \begin{cases} 1, & \text{if } (u,v) \in E \\ 0, & \text{otherwise.} \end{cases} \quad (10)$$

We assume that the likelihood of observing an interaction $u \rightarrow v$ might be influenced by the combined feature values of both $u$ and $v$. Thus, the *independent variable* can be represented as:

$$\mathbf{X}_{u,v} \in \{0,1\}^d \subseteq F \times F \quad (11)$$

where $F$ is the set of all feature values. Specifically, for demographic features, we take $F$ to be $F := \{\text{young, old, male, female, poor, rich}\}$, and $d = |F|^2$. In this context, the logistic regression model is trained on a series of pairs $(y_{u,v}, \mathbf{X}_{u,v})$, producing as output coefficients a feature-feature matrix $M$. This estimated matrix provides insights into which demographic groups have a higher probability of interacting. More explicitly, each matrix entry $a_{ij}$ denotes the log-odds ratio of a node with feature value $i$ to interact with a node with feature value $j$, compared to the probability of random interactions given by the null model.

To define the non-existing edges for the training (where $y_{u,v} = 0$), we employ the configuration model in Section 3.2. By shuffling the existing interaction network edges, we balance positive and negative edge examples. We choose edges based on the activity of the source node $u$ and the attractiveness of the target node $v$ [18]. If such a pair already exists, it is omitted. This method ensures that the estimated matrix $M$ correctly represents how the demographic features make the observed edges deviate from this null model.

**Homophily is evident in demographic features.** Figure 5 reveals pronounced homophily among users based on their demographic features: pairs on the diagonal (i.e., within-class interactions) occur more frequently than expected. Especially the interaction among 'old' users is 18% more likely than expected, and interaction among 'young' and 'female' users is 10% more likely. Figure B.6 shows the same pronounced homophily for /r/PCM, especially interactions among 'young', 'old', and 'poor', which are respectively 15%, 12% and 8% more likely than expected. Conversely, off-diagonal elements show a lower-than-expected interaction probability between users with different demographic features. In Figure B.6 for /r/PCM this effect is particularly notable for the age feature, where 'old' and 'young' interactions are infrequent (odds ratios below 0.90).

## 3.5 Modeling the Combined Effects of Ideological and Demographic Features

Next, we analyze the interplay between ideological and demographic features and their joint effect on user interactions. Additionally, we recognize the potential influence of a user's popularity on Reddit (confounding), which might skew interactions.

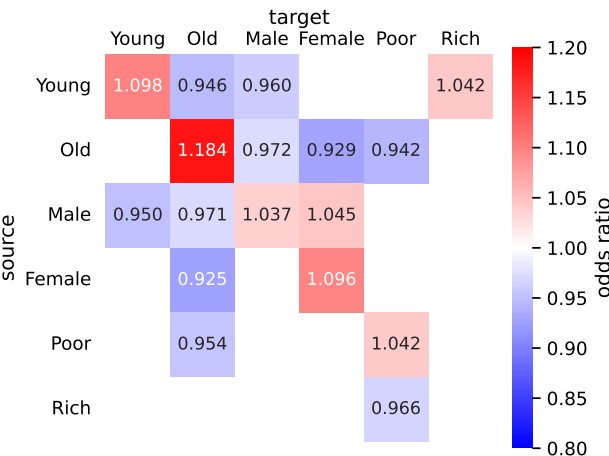

**Figure 5: Odds ratio (exponentiated logistic regression coefficients) for each ordered pair of interacting features on /r/PC. The source user is in the rows and the target user in the columns. Only coefficients significant at the $\alpha = 5\%$ level are shown. The results show homophily in the demographic attributes with higher values in the main diagonal.**

To ensure the significance of our findings, we utilize the logistic regression approach introduced in Section 3.4. Aware of the risk of multicollinearity, where independent variables are highly correlated and can distort results, we opt for a selected combination of ideological and demographic values rather than examining all pairwise combinations. As indicated by Equation (11), the independent dummy variables for our analysis are denoted as $\mathbf{X}_{u,v} \in \{0,1\}^{29}$. The 29 selected features include:

**i.** For ideological features, we consider the following set of pairwise features associated with each source-target edge ($u \rightarrow v$). Given $X_u, X_v \in \{\text{left}, \text{right}\}$, we define

- Economic Homophily = 1 if $X_u = X_v$
- Economic Heterophily = 1 if $X_u \neq X_v$

Given $X_u, X_v \in \{\text{libertarian}, \text{authoritarian}\}$, we define

- Social Homophily = 1 if $X_u = X_v$
- Social Heterophily = 1 if $X_u \neq X_v$

In both cases, users labeled as 'center' do not contribute to the features. The directionality of the interaction is taken into account in a separate feature denoted by an arrow. This design choice allows to separate homophilic/heterophilic effects and their possible asymmetry. Therefore, we consider 6 possible ideologic features: 2 economic features, 2 social features, and 2 features for asymmetry.

**ii.** For demographic features, we consider all the pairwise combinations of values of each feature (age: young-old, gender: male-female, affluence: poor-rich) as those employed to analyze the effect of demographics. Therefore, we have $6 \times (6 + 1)/2 = 21$ possible demographic features.

**iii.** For the confounding effects, we define the popularity of a user on a subreddit as the average score[6] of its comments on that subreddit. Figure A.2 shows the distribution of popularity of

---

[6]A comment's score is the difference between its received upvotes and downvotes.

users for both /r/PC and /r/PCM. Then we quantile-normalize such scores to define 4 classes of popularity, from low to high score, and we consider the top and bottom quartiles as two distinct binary features. By taking into account the target's popularity, we define

- Target is popular = 1 if the target is in the top quartile
- Target is not popular = 1 if it is in the bottom quartile

We thus add 2 confounding features for popularity.

Figure 6 reports the results of the logistic regression model.

**Heterophily in economic ideologies.** Having opposite economic ideologies increments the odds of an interaction by 10% in /r/PC (Figure 6) and higher than 16% in /r/PCM (Figure B.7). This effect is mirrored by a decrement of the odds by 5% if both users have the same economic ideology for /r/PC, and by 12% for /r/PCM. Results for directional heterophily are not statistically significant both in /r/PC and /r/PCM.

**Homophily in social ideologies.** Conversely, pairs of users with the same social ideology are more likely to interact, with an increment of the odds of 8% for /r/PC (Figure 6) and slightly higher than 5% for /r/PCM (Figure B.7). Similarly, heterophilic interactions across different social ideologies are less likely by approximately 7% for /r/PC and slightly higher than 5% for /r/PCM. Results for directional heterophily are not statistically significant both in /r/PC and /r/PCM.

**Homophily in demographics.** Age homophily significantly predicts interactions, with a 13% increase in odds for old-old pairs and a 6% increase for young-young pairs in /r/PC (Figure 6). In /r/PCM (Figure B.7), these odds increase by 13% and 14%, respectively. Demographic homophily in terms of poor-poor, female-female, and male-male interactions shows higher likelihood, with odds increasing by 8%, 7%, and 5% for /r/PC and 8%, 4%, and 2% for /r/PCM. Results for rich-rich interactions are not statistically significant for /r/PC. However, in /r/PCM, they cause an odds increment of 4%.

**No heterophily in demographics.** Generally, heterophily in demographics reduces the odds of interactions. In particular, poor-rich and young-old interactions in /r/PC decrease odds both by 4% (Figure 6). In /r/PCM (Figure B.7), these reductions are 5% and 11%.

Furthermore, cross-feature interactions generate different results. Old-poor and old-female interactions in /r/PC decrease odds by 4% and 10% (Figure 6). In /r/PCM (Figure B.7), these reductions are both of 4%. Old-rich interactions cause a 3% increase in odds for both /r/PC and /r/PCM. Female-rich and young-rich interactions yield opposite results, with odds increasing in /r/PC and decreasing in /r/PCM. Old-male and young-male interactions lead to a decrease in odds for /r/PC but are not statistically significant for /r/PCM. Young-female interactions are not statistically significant for /r/PC but result in an odds increment for /r/PCM.

## 4 TOXICITY IN SOCIAL INTERACTION

We found a higher rate of heterophilic interactions on the economic axis of the political compass. Here we show that heterophilic interactions are in general characterized by higher *toxicity*, which hints at conflictual interactions between axes poles. Toxicity is defined as "a rude, disrespectful, or unreasonable comment that is likely to make you leave a discussion".[7]

---

[7]https://developers.perspectiveapi.com/s/about-the-api-attributes-and-languages

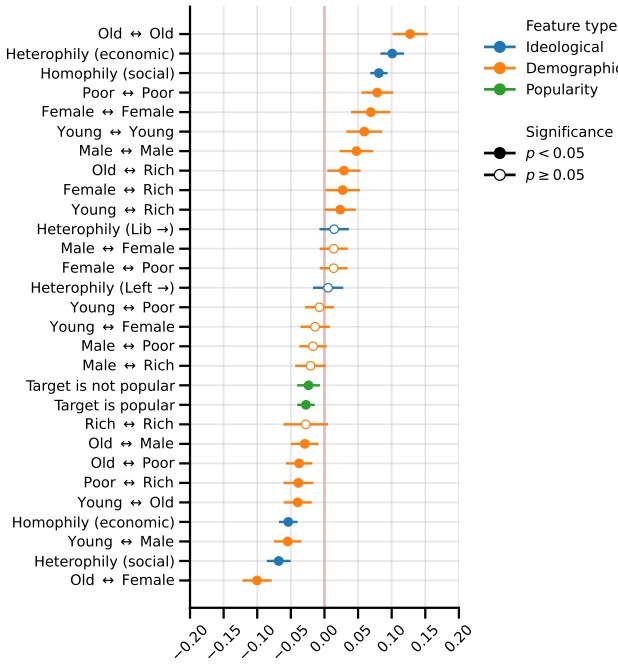

**Figure 6: Coefficients and 99% confidence intervals for logistic regression features on /r/PC. The features are displayed in the rows and they represent pairs of classes in ideological characteristics (blue), demographic characteristics (orange), or popularity (green). A coefficient greater than 0 means a positive impact on the likelihood of interaction. Statistical significant results are highlighted with full markers.**

**Estimating language toxicity.** We use Google's Perspective API[8] to determine the toxicity scores of a sample of 100k comments from both /r/PC and /r/PCM, excluding comments containing only emojis or links. A toxicity score $\tau$ ranges between 0 (lowest) and 1 (highest), and indicates the likelihood that an individual perceives the text as toxic. For each pair of interacting ideologies $(X, Y)$, $\overline{\tau}_{X \rightarrow Y}$ denotes the average toxicity of the comments exchanged.

We then compare this empirical measure against the one obtained in a null model of interactions that preserves the degree sequences and the toxicity distribution. We sample 100 randomized interaction networks from the null model for each subreddit via an MCMC edge-swapping algorithm [27]. Randomization is achieved by executing $Q \cdot |E|$ edge endpoint swaps, with $|E|$ being the edge count of the network. Although there is no definite proof for Markov Chain convergence, we adhere to a conservative value of $Q = \log|E|$, as recommended by Uzzi et al. [61].

**Heterophilic interactions show higher toxicity.** Figure 7 depicts the ratio of the average empirical toxicity to the average toxicity in the null model for /r/PC. On both axes, heterophilic comments present a higher toxicity than expected, while homophilic comments present a lower one (on-diagonal values are lower than 1, while off-diagonal ones are higher than 1). On the economic axis, heterophilic comments (i.e., between left- and right-leaning

---

[8]https://perspectiveapi.com

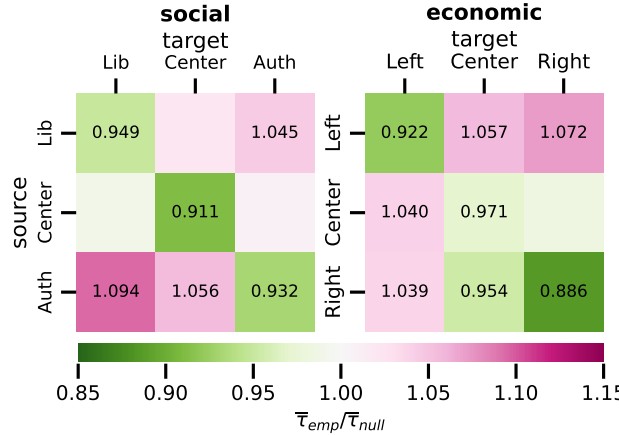

**Figure 7: Ratio between the average toxicity in empirical data ($\overline{\tau}_{\text{emp}}$) and random null model ($\overline{\tau}_{\text{null}}$) for /r/PC. Values above 1 indicate higher toxicity than expected. Non-significant results at the $\alpha = 5\%$ level are omitted (via a t-test). The left heatmap shows interactions between ideologies on the social axis, and the right heatmap focuses on the economic axis. Heterophilic interactions (lib-auth, left-right) exhibit higher toxicity than expected.**

users) are 4-to-7% more toxic than expected, and statistically significant in both directions (p-values of 0.0004 and $< 10^{-6}$). Conversely, interactions between users of the same ideology demonstrate lower-than-expected toxicity with an effect of 8-to-12% (p-values $< 10^{-6}$).

Regarding the social axis, homophilic comments between users of identical political ideologies are 5-to-9% less toxic than expected (p-values $< 10^{-6}$). Instead, comments from authoritarians towards libertarians are significantly more toxic (10%, p-value $< 10^{-6}$), with a similar albeit smaller effect in the opposite direction.

People who interact with others who have different political views (Figure 4) are more likely to engage in toxic behavior, while people who interact with others who have the same political views are less likely to engage in toxic behavior.

The same considerations apply to /r/PCM, but the results are less evident and/or not statistically significant, as shown in Figure B.8. This suggests that a more humorous discussion involving memes, as those in /r/PCM, may be less conflictual than a serious discussion, such as those in /r/PC.

## 5 DISCUSSION

In our quantitative analysis of interaction patterns on Reddit's communities /r/PoliticalCompass and /r/PoliticalCompassMemes from 2020 to 2022, we uncovered dynamics that go beyond the conventional left–right political dichotomy. Users with similar social ideologies and demographics interacted more frequently and typically used non-toxic language, which denotes a high level of homophily on the social axis of the political compass. Instead, conversations reflecting "social" heterophily, particularly between authoritarian and libertarian ideologies, occurred 6% less frequently than expected. In contrast, heterophilic and conflictual interactions were pronounced on the economic axis, with users of opposing economic ideologies displaying higher signs of toxicity and engaging with each other 10% more than expected.

In light of our results, some of the apparently puzzling differences in the recent literature may be reconciled. On the one hand, social media platforms such as Twitter and Facebook, which emphasize social connections, reportedly exhibit a high level of homophily [17]. The presence of political echo chambers [2, 13, 32] is therefore to be expected if this social homophily is a driver of the interactions, as suggested by our results. On the other hand, platforms such as Reddit, where status homophily is less dominant and interactions are centered on shared topical interests, tend to experience more conflictual interactions across ideologies [60]. This fact is particularly true for Reddit's political spaces, which skew towards the U.S., a nation where the political spectrum largely corresponds to a singular economic left-right dimension [37]. As a result, echo chambers may be rarer on platforms like Reddit [13, 18, 46]. Building on our findings, it is evident that the nature of interactions, in terms of toxicity along with the specific ideological axis, significantly influences the manifestation of homophily and heterophily.

Our findings also highlight the inclination of users to group based on demographics and similar social ideologies, hinting at a certain degree of social affinity [62]. Belonging to such digital echo chambers provides a sense of community but also poses risks. Being predominantly exposed to only one type of ideology can fuel misinformation [20, 59]. It can strengthen existing biases and further spread incorrect beliefs. If these online tendencies continue, they might intensify real-world divisions with consequences in voting patterns and everyday interactions [53].

**Limitations and future work.** Our study is not exempt from limitations. Firstly, our dataset lacks geographical granularity; it focuses on English-speaking users predominantly from a U.S.-centric platform, Reddit. Future studies should aim to infer and incorporate geographical locations of users to control for regional effects. Secondly, our reliance on self-declarations means that we excluded users who change their political orientation in time. An interesting direction for future research is analyzing such opinion changes, under the lens of the political discussions that occurred within the subreddit and outside of it.

Moreover, user declarations may not strictly align with the true ideology of users nor with the political content in the observed subreddits. While declared ideologies along the economic axis are validated by comparison to left/right ideologies inferred by following Waller and Anderson [63], we could not validate self-declarations along the social axis. Future work could be devoted to building a more comprehensive embedding of Reddit users that takes into account their political stance in a multidimensional way, e.g., by adding the libertarian–authoritarian axis. Likewise, the textual content of the interactions is a rich resource that can yield further insights and has not yet been fully utilized.

Lastly, our findings pave the way for multidimensional modeling and intervention studies. For instance, a social compass model has been recently proposed to explore depolarization dynamics in multidimensional topics represented in a polar space [48, 49]. Similarly interesting directions for future work are the exploration of algorithms that offer diverse content by taking into account the multidimensional nature of targeted users [30, 31].

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

# A DATASET DESCRIPTION

This Appendix provides additional information regarding the data sets used in this work, from /r/PC and /r/PCM. Table A.1 reports the fractions of deleted users and users with unique or non-unique political flairs in both subreddits. Figure A.1 shows the number of posts and comments in the years from 2018 until 2022 in both subreddits. Figure A.2 shows the probability density function of activity and popularity of users in both subreddits.

**Table A.1: Fraction of posts and comments by user category in /r/PC and /r/PCM. The rows represent the following user types, in order: deleted users, bots (as listed in [54]), users lacking a unique political declaration, and users with a unique political declaration. The last row indicates the fraction of analyzed data.**

| Subreddit | /r/PC | | /r/PCM | |
|---|---|---|---|---|
| User Type | Posts | Comments | Posts | Comments |
| Deleted | 34.72% | 9.49% | 30.25% | 10.82% |
| Bots | 0.25% | 0.6% | 0.73% | 2.57% |
| Non-unique political flairs | 28.41% | 30.83% | 21.98% | 22.28% |
| Unique political flairs | **36.62%** | **59.08%** | **47.04%** | **64.33%** |

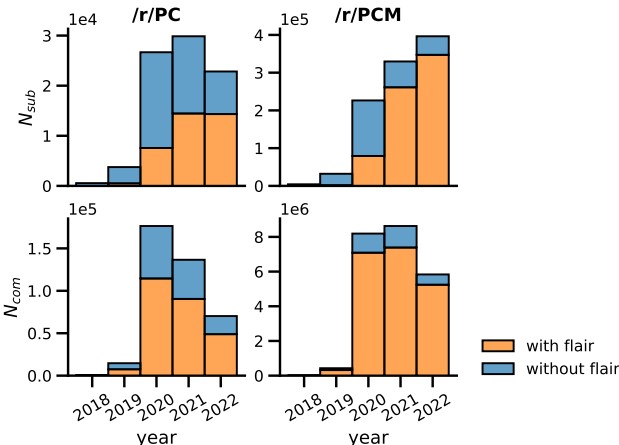

**Figure A.1: Number of posts and comments over time in both subreddits.**

# B ADDITIONAL FIGURES FOR /R/POLITICALCOMPASSMEMES

This Appendix shows additional figures for /r/PCM, analogous to the ones presented in the main text. In all cases, results are qualitetively and quantitatively similar to /r/PC.

Figure B.3 shows the number of users and comments grouped by political ideology for /r/PCM, equivalent to Fig. 2 in the main text for /r/PC. Figure B.4 shows the mean and confidence intervals of quantiles for each socio-demographic feature with respect to ideologies for /r/PCM. Figure B.5 shows the odds ratios between

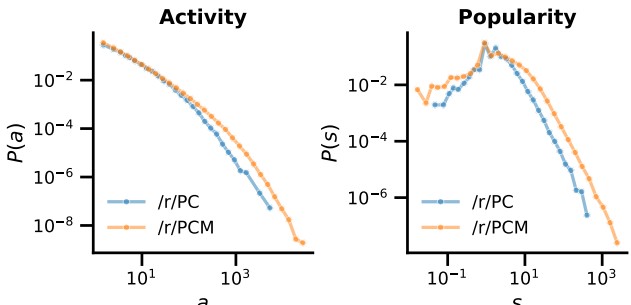

**Figure A.2: Activity ($a$) and popularity (average score $s$) distributions for both subreddits.**

empirical and random conditional probabilities of interaction for /r/PCM, equivalent to Figure 4 in the main text for /r/PC. Figure B.6 shows the odds ratios for each ordered pair of demographic features for /r/PCM, equivalent to Figure 5 in the main text for /r/PC. Figure B.7 shows the the coefficients and confidence intervals for logistic regression features for /r/PC, equivalent to Figure 6 in the main text for /r/PC. Figure B.8 shows the ratio between average toxicity in empirical data and random null model for /r/PC, equivalent to Figure 7 in the main text for /r/PC.

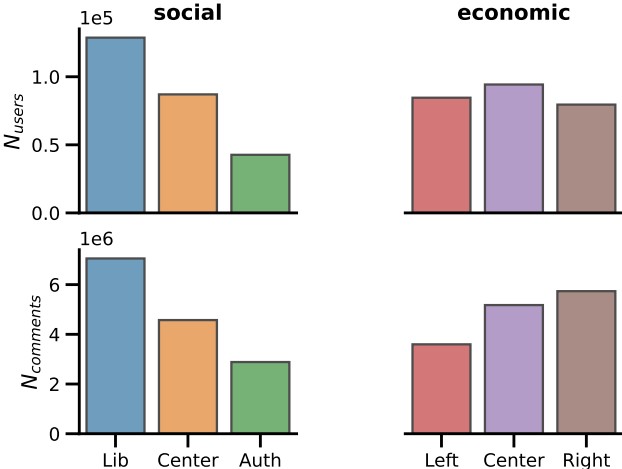

**Figure B.3: Composition of /r/PCM in terms of the number of users and comments classified by political ideology.**

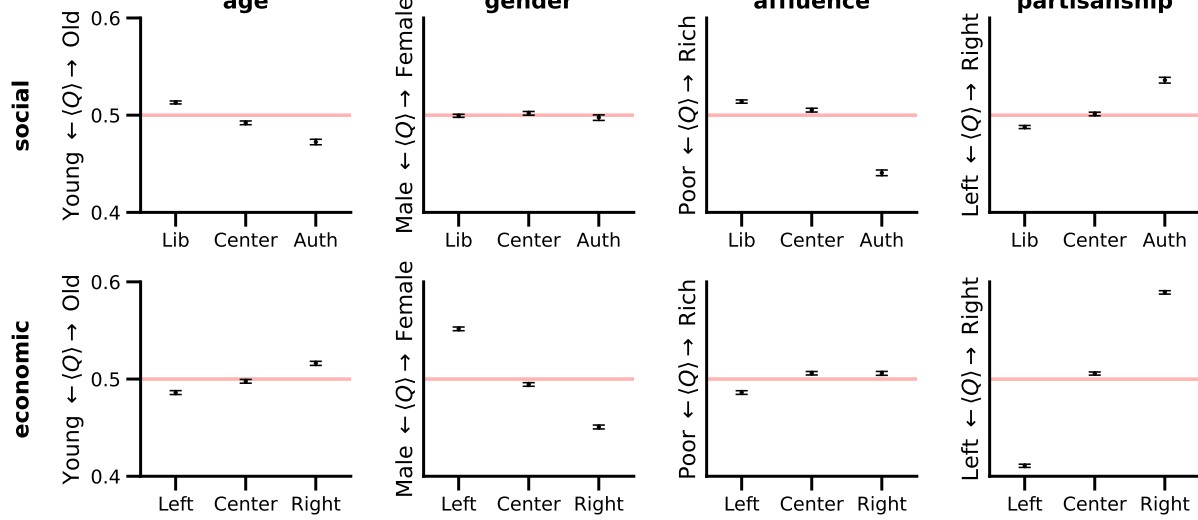

**Figure B.4: For each axis (social and economic) displayed in the rows, and for each user characteristic (age, gender, affluence, partisanship) displayed in the columns, the plot shows the average quantile score and its 95% confidence intervals for /r/PCM. Libertarians tend to be older, while right-leaning users are predominantly male. Left-leaning users are more likely to be female. Additionally, a correlation between partisanship and the left–right economic axis emerges, which further validates our data collection methodology.**

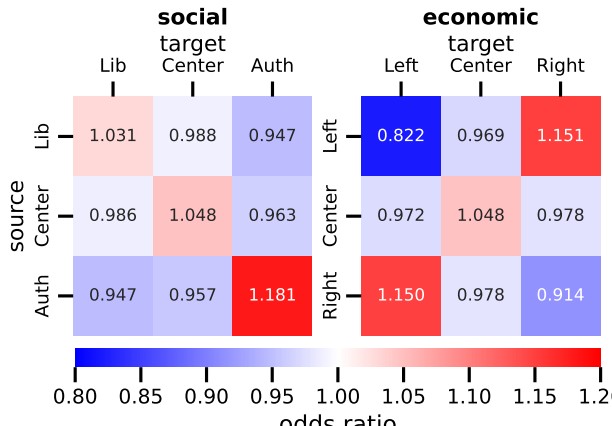

**Figure B.5: Odds ratios between empirical and random conditional probabilities of interaction for /r/PCM, with respect to social (left) and economic (right) axes. The interactions show a homophilic pattern on the social axis (higher values in the main diagonal) and a heterophilic one on the economic axis (higher values in the anti-diagonal).**

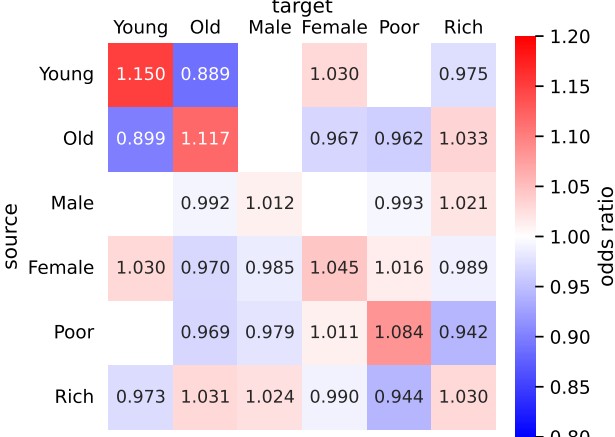

**Figure B.6: Odds ratio (exponentiated logistic regression coefficients) for each ordered pair of interacting features on /r/PCM. The source user is in the rows and the target user in the columns. Only coefficients significant at the $\alpha = 5\%$ level are shown. The results show homophily in the demographic attributes with higher values in the main diagonal.**

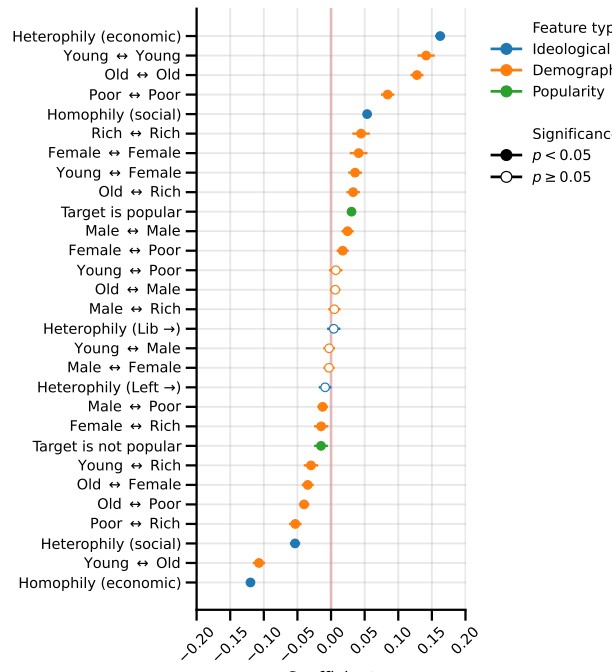

Figure B.7: Coefficients and 99% confidence intervals for logistic regression features on `/r/PCM`. The features are displayed in the rows and they represent pairs of classes in ideological characteristics (blue), demographic characteristics (orange), or popularity (green). A coefficient greater than 0 means a positive impact on the likelihood of interaction. Statistical significant results are highlighted with full markers.

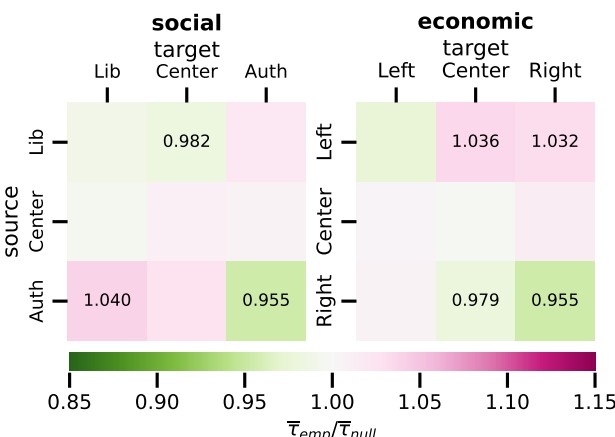

Figure B.8: Ratio between the average toxicity in empirical data ($\overline{\tau}_{\mathrm{emp}}$) and random null model ($\overline{\tau}_{\mathrm{null}}$) for `/r/PCM`. Values above 1 indicate higher toxicity than expected. Non-significant results at the $\alpha = 5\%$ level are omitted (via a t-test). The left heatmap shows interactions between ideologies on the social axis, and the right heatmap focuses on the economic axis. Heterophilic interactions (Lib-Auth, Left-Right) exhibit higher toxicity.

