# OpenReview forum: "Navigating Multidimensional Ideologies with Reddit’s Political Compass: Economic Conflict and Social Affinity"
_ACM.org/TheWebConf/2024/Conference — TheWebConf24 Oral_

### Official Review · Reviewer_6uWa · 2023-11-19

**Novelty:** 4
**Technical Quality:** 4

**Review:**

The paper investigates online political discourse using a multidimensional ideological framework, examining over 8 million comments from Reddit between 2020 and 2022. The study dissects users' ideological dimensions into economic (left-right) and social (libertarian-authoritarian) axes, and correlates them with demographic attributes. The research uncovers significant homophily in social interactions along the social axis and demographic characteristics, while revealing heterophily along the economic axis. Additionally, the paper examines the language toxicity in these interactions, revealing higher toxicity in heterophilic interactions, indicating a more conflictual discourse.

**Advantages**

S1. The paper proposes a novel model for analyzing comments by disentangling users' ideological dimensions into economic (left-right) and social (libertarian-authoritarian) axes.

S2. The paper analyzes a large real-life dataset, consisting of 8 million comments, allowing for a comprehensive exploration of the political landscape on Reddit.

S3. The paper presents interesting findings from the dataset mining, such as significant homophily in social interactions along the social axis and demographic characteristics, while revealing heterophily along the economic axis.

**Disadvantages**

W1. The paper solely focuses on datasets from Reddit, and only considers users from the US, which may limit the generalizability of the findings to other social media platforms.

W2. The dataset used in the paper relies on users' self-declared political affiliations, which may not always be accurate or sincere.

**Questions:**

Q1. It requires further discussion on whether the datasets from Reddit are representative and if the findings and conclusions from the paper are universally applicable.

Q2. It is necessary to discuss how the issue mentioned in W2 might impact the conclusions drawn in the paper.

**Reviewer Confidence:**

2: The reviewer is willing to defend the evaluation, but it is likely that the reviewer did not understand parts of the paper

**Scope:**

4: The work is relevant to the Web and to the track, and is of broad interest to the community

---

### Official Review · Reviewer_NMYK · 2023-11-22

**Novelty:** 6
**Technical Quality:** 7

**Review:**

**Paper summary**

This paper explores the dynamics of opinion formation in populations, particularly focusing on opinion polarization, homophily, heterophily, and its manifestations in social media interactions. It also delves into the multidimensional nature of opinion dynamics, considering various ideological axes and demographic factors, and presents empirical observations from Reddit interactions along these axes.

**Strengths**

1. This paper provides a comprehensive analysis of social interactions on Reddit, investigating how users' ideological positions align with their demographics and how these interactions manifest along different ideological dimensions. I especially like that the authors provided external validations on some of the measures

2. The paper addresses the apparent contradictions in the existing literature by examining both homophily and heterophily in political discussions, shedding light on the nuanced nature of these interactions.

3. Methodologies used are robust and well justified. The text effectively communicates the key findings, emphasizing the prevalence of homophily in interactions based on social ideologies and demographics, contrasting with more conflictual interactions along the economic axis.

**Questions:**

**Minor suggestions**
1 . Conflictual interactions could also be represented by controversiality or scores. It would be interesting to see if high toxicity correlations between different ideologies align with the low comment scores

2. The discussion could benefit from some writing on how the methods used can be ported over other social media platforms.

**Ethics Review Description:**

No ethical flags

**Reviewer Confidence:**

3: The reviewer is confident but not certain that the evaluation is correct

**Scope:**

4: The work is relevant to the Web and to the track, and is of broad interest to the community

---

### Official Review · Reviewer_oegi · 2023-11-24

**Novelty:** 7
**Technical Quality:** 7

**Review:**

In this study, the authors analyze 8 million comments from various subreddits over two years (2020-2022), focusing on users' self-declared positions on economic and social axes. Utilizing network analysis and a socio-demographic model, they discovered notable homophilic interactions among users with similar social ideologies and significant heterophilic interactions among those with differing economic ideologies. A higher level of toxicity is associated with heterophilic interactions. These findings offer insights into the mixed results of homophilic and heterophilic interactions on social media and contribute significantly to research on multidimensional ideologies online.

**Questions:**

While the nature of Reddit might diminish homophilic interactions or echo chambers, the heightened toxicity in economic ideology-based conflictual interactions could actually intensify polarization. This result challenges theories that suggest the potential for deliberative discussion when users are exposed to contrasting viewpoints. Incorporating a discussion about how these findings relate to theories of deliberative discourse could significantly strengthen the manuscript's contribution. Such an analysis would provide a nuanced understanding of how exposure to differing viewpoints on platforms like Reddit might not always facilitate constructive debate, particularly in the context of economic ideologies.

**Reviewer Confidence:**

3: The reviewer is confident but not certain that the evaluation is correct

**Scope:**

4: The work is relevant to the Web and to the track, and is of broad interest to the community

---

### Official Review · Reviewer_bpMu · 2023-11-24

**Novelty:** 5
**Technical Quality:** 3

**Review:**

This study analyzes social interaction on Reddit, focused on two subreddits /r/PoliticalCompass and /r/PoliticalCompassMemes. Relying on self-reports of two ideological dimensions (economic and social), they find homophilic interactions along the economic axis but the opposite along the social axis. Finally, they find that heterophilic interactions are more toxic.

Pros
* Theoretical motivations (though it is on an extensively studied topic) are clear, and the paper's objectives are well defined
* Use of self-reported ideology measures is ingenious
* Analysis spanning Sections 3-4 are quite convincing

Cons
* The biggest issue is sampling bias. What kind of people are the users under study (those who on Reddit and self-report their ideology?) It is not difficult to how unrepresentative they are neither of the general population of course nor of social media users (or even Reddit users). While the authors discuss the limitations of using self-reports, those limitations are relatively minor in my view. The users under study, some certain unobserved characteristics, like political sophistication or political interest, can shape patterns and tone of homophilic and heterophilic interactions.
* The finding on toxicity of cross-cutting discussions is a bit unsurprising

**Questions:**

See above please.

**Ethics Review Description:**

-

**Reviewer Confidence:**

4: The reviewer is certain that the evaluation is correct and very familiar with the relevant literature

**Scope:**

4: The work is relevant to the Web and to the track, and is of broad interest to the community

---

### Decision · Program_Chairs · 2024-01-22

**Decision:**

Accept (Oral)

**Comment:**

The reviewers are generally positive about this paper and I am happy to recommend its acceptance with the condition that the authors address some of the points raised by reviewers:
 - Questions around sampling bias - please make sure to discuss this in the paper upfront (in the Method section) as well towards the end (Discussion section) to elaborate what could be alternative ways of studying this space so as to mitigate the sampling bias
 - Generalizability - Multiple reviewers have pointed out the issue with generalizability and authors have minimal attempt to address this concern. While I agree that the scope of the paper is the study of these two specific communities, I think its worthwhile to discuss how the findings can be potentially useful in the context of other platforms and other countries (beyond the US). The rebuttal response says - "our study paved the way to future research on disentagling political polarization along economic and social dimensions on other platforms, across countries", but it fails to mention any further details. The paper needs to have a future work section elaborating on these points.